# Artificial Intelligence-Based Multimodal Medical Image Fusion Using Hybrid S$^2$ Optimal CNN

**Marwah Mohammad Almasri [1,\*] and Abrar Mohammed Alajlan [2]**

[1] Department of Computer Science, College of Computing and Informatics, Saudi Electronic University, Riyadh 11673, Saudi Arabia
[2] Self Development Skills Department, Common First Year Deanship, King Saud University, Riyadh 11451, Saudi Arabia; alajlan1@ksu.edu.sa
[\*] Correspondence: m.almasri@seu.edu.sa

**Abstract:** In medical applications, medical image fusion methods are capable of fusing the medical images from various morphologies to obtain a reliable medical diagnosis. A single modality image cannot provide sufficient information for an exact diagnosis. Hence, an efficient multimodal medical image fusion-based artificial intelligence model is proposed in this paper. Initially, the multimodal medical images are obtained for an effective fusion process by using a modified discrete wavelet transform (MDWT) thereby attaining an image with high visual clarity. Then, the fused images are classified as malignant or benign using the proposed convolutional neural network-based hybrid optimization dynamic algorithm (CNN-HOD). To enhance the weight function and classification accuracy of the CNN, a hybrid optimization dynamic algorithm (HOD) is proposed. The HOD is the integration of the sailfish optimizer algorithm and seagull optimization algorithm. Here, the seagull optimizer algorithm replaces the migration operation toobtain the optimal location. The experimental analysis is carried out and acquired with standard deviation (58%), average gradient (88%), and fusion factor (73%) compared with the other approaches. The experimental results demonstrate that the proposed approach performs better than other approaches and offers high-quality fused images for an accurate diagnosis.

**Keywords:** multimodality image fusion; artificial intelligence; discrete wavelet transform; cnn; optimization

## 1. Introduction

Image fusion represents the procedure of combining the image diversity obtained by the different modalities. It is widely utilized in the diagnosis of the disease, surgery, and treatment planning. The various classes such as bones, organs, or tissues are reflected in several medical images with several modalities. The fusion process in medical application is utilized for image correctness and detection as well as assessment of medical issues by preserving and enlightening the relevant attributes [1]. In the medical field, acomputed tomography (CT) image observes thick frameworks such as bones when compared to magnetic resonance imaging (MRI) in the break examination. MR imaging provides information linked to the soft tissues for reflecting the absorbed information, progression, such as single-photon emission CT (SPECT), and positron emission tomography (PET). PET offers highly sensitive images and SPECT represents the nuclear imaging method used for exploring the flow of blood in organs as well as tissues [2]. The major application of this fusion is for extracting the medical information from various sensors that is normally not visible in the image form. Some biomedical sensors like ultrasound, PET, MRI, X-ray, and CT provide clinical information usingthe reflection of the human body organs [3]. To obtain proper information regarding perfect detection, clinicians are usually required to merge various kinds of medical images from an identical location to detect the causes

of a patient's issues. Image fusion methods offer an efficient scheme for resolving these problems. Medical image fusion methods fuse the multi-modality medical images for accurate as well as reliable medical detection [4].

Image fusion is classified into three levels, namely, feature-level, decision level and pixel-level. Feature level fusion realizes the feature specifications and their dissimilarities such as color, shape, texture, edge, etc. and integrates the dissimilarities that depend upon the feature resemblance [5]. In feature level fusion, the features are removed distinctly from every source image. Decision level fusion is utilized to combine the higher-level outcomes from the various algorithms to obtain the final decision for the fusion procedures. Every image is first fused independently and then provided to the fusion process. Decision level fusion divides the pixels from the various source images, which depends upon the extracted features, and gets the decision for the suitable class label for every pixel [6]. Pixel level fusion is utilized to conserve the spatial features of the source image pixels. Hence, numerous pixel level fusion techniques have been offered recently. Pixel-level fusion is classified into two types of methods depending on their modes, such as the transform domain-based and the spatial domain-based image fusion methods. However, the image fusion is developed through both the transform domain-based as well as the spatial domain-based methods [7].

Spatial domain-based fusion methods utilize local features such as the standard deviation, spatial frequency, and gradient of the source images. The normally utilized techniques in the spatial domain methods contain intensity hue saturation (IHS) and principal component analysis (PCA). The fused images achieved via these techniques generally suffer from high spatial distortion and low SNR. In the transform domain schemes, the source images are decomposed into expressive sub-bands to distinguish the salient attributes such as edges and sharpness of the image. The standard transform domain fusion methods depend upon multi-resolution analysis (MRA) [8–10]. In medical applications, medical image fusion methods are capable of fusing the medical images from various morphologies to obtain reliable medical diagnosis. The single modality image cannot provide sufficient information for exact diagnosis. Hence, this paper proposes multimodality medical image fusion based on a CNN with a hybrid optimization dynamic (HOD) algorithm in the discrete wavelet transform. Initially, the multimodal medical images are transmitted into the MDWT and optimization models are utilized to obtain the fused images. The fused images are then classified into malignant or benign using a CNN-HOD classifier. The main purpose of the HOD algorithm is to improve classification accuracy. The experimental results reveal that the proposed method performs better than the existing multimodality medical image fusion methods. The major contribution of the paper is as follows.

A modified discrete wavelet transform (MDWT) is utilized to decompose the images into low- and high-frequency sub-bands.

The fused images are classified as malignant or benign using the proposed convolutional neural network-based hybrid optimization dynamic algorithm (CNN-HOD).

The proposed approach is compared with various other image fusion-based techniques to evaluate the performance of the system.

## 2. Review of Biomedical Imaging Process

Yadav et al. [11] proposed the hybrid discrete wavelet transform and principal component analysis (PCA) techniques (DWT-PCA) for the process of medical image fusion using image modalities such as MRI, PET, SPECT, and CT. Poor image quality and inconsistent performances with minimum efficiency was considered as the significant drawback of this approach. Subbiah et al. [12] proposed the enhanced monarch butterfly optimization algorithm and discrete shearlet transform with restricted Boltzmann machine (EMBO-DST with RBM) approach for multimodal medical image fusion. The medical image fusion was achieved using four sets of benchmark database images (represented as D1, D2, D3, and D4), consisting of the MRI, PET, SPECT, and CT images. This technique faced a few difficulties during the implementation process and failed to perform under real-time applications.



Wang et al. [13] proposed a convolutional neural network for fusing the pixel activity information of input source images to understand the creation of weight maps. Eight various image fusion methods were utilized, fusing images such as MRI, CT, T1, T2, PET, and SPECT. The major drawback of this method was that it was difficult to fuse infrared-visible and multi-focus image fusion. Parvathy et al. [14] proposed the discrete gravitational search algorithm (DGSA) with a deep neural network to improve the classification accuracy. The proposed method utilizes four datasets (I, II, III, and IV) that include the modalities such as CT, SPECT, and MRI. The performance of the proposed method was evaluated using measures such as sensitivity, accuracy, precision, specificity, fusion factor, and spatial frequency. The major difficulty of this method was implementing it in real-time applications.

Tan et al. [15] proposed a pulse coupled neural network in a non-subsampled shearlet transform approach to improve the fusion quality of medical images. Above 100 pairs of multimodal medical images were obtained from the Whole Brain Atlas dataset in which the modalities consist of MRI, PET, and SPECT. Li et al. The authors of [16] proposed a Laplacian re-decomposition method (LRD) to enhance the multimodal medical image fusion quality. This method utilized 20 pairs of multimodal medical images collected from Harvard University medical library. Arif et al. [17] proposed a novel method for fusion of multimodal medical images that depends on curvelet transform as well as a genetic algorithm (GA). The dataset was achieved at CMH Hospital Rawalpindi from modalities such as CT, MRI, PET, MRA, and SPECT. The major drawback of this method was determining the decomposition level. Kaur et al. [18] decompose an image using non-subsampled contourlet transform with multi-objective exception as well as differential evolution for the multimodality medical image fusion. The major drawback of this method was fusing the remote sensing images.

Hu et al. [19] proposed anew fusion method that integrates separable dictionary optimization with a Gabor filter in the non-subsampled contourlet transform (NSCT) domain. The proposed method was tested on 127 groups of brain anatomical images from the Whole Brain Atlas medical image database with modalities such as MRI and CT images. The major drawback of this method was the greater time consumption. Xia et al. [20] proposed a parameter-adaptive pulse-coupled neural network (PAPCNN) method to obtain a better fusion effect. The proposed method utilized 70 pairs of source images collected from the Whole Brain Atlas of Harvard Medical School [13] and the Cancer Imaging Archive (TCIA). The medical images were fused using modalities such as CT, MRI, T1, T2, PET, and SPECT. Table 1 depicts a summary of related works on multimodality medical image fusion.

Shehanaz et al. [21] suggested optimum weighted average fusion (OWAF) with a particle swarm algorithm (PSO) to enhance the performance of multimodal mapping. The multi-modality imaging pairs, namely, MR-CT, MR-SPECT, and MR-PET, were used for the evaluation of the OWAF method. The simulation setup was carried out using a public image dataset that contains normal and diseased brain images (http://www.med.harvard.edu/AANLIB/, accessed on 18 January 2022). The result showed that the OWAF-PSO method achieved greater fusion qualities, but it took more computational time to perform the task.

To enhance the quality of fusion images, Dinh et al. [22] introduced a sum of local energy function with a Prewitt compass operator (SLE-PCO) along with an equilibrium optimizer algorithm (EOA). In this, SLE-PCO increases the contrast of the image and EOA prevents the loss of significant data. The performance of this approach was validated using MRI-PET medical images taken from the website http://www.med.harvard.edu/AANLIB/ (accessed on 18 January 2022). This approach efficiently improves low contrast medical images and conserves detailed layers of data, but the drawback was high computational complexity.

**Table 1.** Summary of related works regarding multimodal medical image fusion.

| Authors | Fusion Schemes | Modality | Datasets | Metrics | Cons |
|---|---|---|---|---|---|
| Yadav et al. [11] | Hybrid DWT-PCA | MRI, PET, SPECT, CT | Online repository datasets | EN, SD, RMSE, and PSNR | Low image quality, performance was not consistent so low efficiency |
| Subbiah et al. [12] | EMBO-DST with RBM model | MRI, PET, SPECT, CT | Four sets of database images (represented as D1, D2, D3, and D4) | SD, EQ, MI, FF, EN, CF and SF | Implementation was complex |
| Wang et al. [13] | CNN | MRI, CT, MRI, T1, T2, PET, and SPECT | Online eight fused images | TE, AB/F, MI, and VIF | Difficult to fuse infrared-visible and multi-focus image fusion. |
| Parvathy et al. [14] | DNN with DGSA | CT, SPECT, and MRI | Four datasets (I, II, III and IV) | Fusion factor and spatial frequency | Failed to execute in real-time applications |
| Tan et al. [15] | PCNN-NSST | MRI, PET, and SPECT | 100 pairs of multimodal medical images from the Whole Brain Atlas dataset | Entropy (EN), standard deviation (SD), normalized mutual information (NMI), Piella's structure similarity (SS), and visual information fidelity (VIF) | Fused image quality was poor |
| Li et al. [16] | Laplacian re-decomposition method (LRD) | MRI, PET, and SPECT | 20 pairs of multimodal medical images collected from Harvard University medical library | Standard deviation (STD), mutual information (MI), universal quality index (UQI), and tone-mapped image quality index (TMQI) | Difficult to propose more rapid and active methods of medical image enhancement and fusion |
| Kaur et al. [18] | NSCT | MRI, CT | Multi-modality biomedical images dataset is obtained from Ullah et al. (2020) [5] | Fusion factor, fusion symmetry, mutual information, edge strength | Difficult to fuse the remote sensing images |
| Hu et al. [19] | Analytic separable dictionary learning (ASeDiL) method in NSCT domain | CT and MRI | 127 groups of brain anatomical images from the Whole Brain Atlas medical image database | Piella–Heijmans' similarity based metric $QE$, spatial frequency (SF), universal image quality index (UIQI), and mutual information | Time consumption was more |
| Xia et al. [20] | Parameter-adaptive pulse-coupled neural network (PAPCNN) method | CT, MRI, T1, T2, PET, and SPECT | Database from the Whole Brain Atlas of Harvard Medical School [13] and the Cancer Imaging Archive (TCIA) | Entropy (EN), edge information retention (QAB/F), mutual information (MI), average gradient (AG), space frequency (SF), and standard deviation (SD) | Implementation was complex |

**Table 1.** *Cont.*

| Authors | Fusion Schemes | Modality | Datasets | Metrics | Cons |
|---|---|---|---|---|---|
| Shehanaz et al. [21] | Optimum weighted average fusion (OWAF) with particle swarm algorithm (PSO) | MR-CT, MR-SPECT, and MR-PET | Brain images (http://www.med.harvard.edu/AANLIB/) accessed on 18 January 2022 | Standard deviation (STD), mutual information (MI), universal quality index (UQI), | Required more computational time to perform the task |
| Dinh et al. [22] | SLE-PCO with EOA | MRI-PET medical images | http://www.med.harvard.edu/AANLIB/ accessed on 18 January 2022 | SD, EQ, MI, FF, EN, CF, and SF | Computational complexity was high |
| Dinh et al. [23] | FRKCO and MPA | MRI-PET medical images | http://www.med.harvard.edu/AANLIB/ accessed on 18 January 2022 | Standard deviation (STD), mutual information (MI), universal quality index (UQI), and tone-mapped image quality index (TMQI) | Information entropy was low |

Dinh et al. [23] demonstrated a three-scale decomposition (TSD) technique, a rule base on local energy function using a Kirsch compass operator (FR-KCO) and a marine predators algorithm (MPA) to enhance image details, preserve significant data, and increase image quality, respectively. The MRI-PET medical images were utilized from http://www.med.harvard.edu/AANLIB/ (accessed on 18 January 2022) to determine the performance of the approach. This method achieved higher fusion performance while the limitation was low information entropy.

## 3. Proposed Methodology

The block schematic of the proposed multimodality medical image fusion is depicted in Figure 1. The information source is fused from a single source of various interval times. In the first stage, the image is fused by deliberating the modified discrete wavelet transform. The input images are CT, MRI, PET, and SPECT images. These images are obtained from online or near the scan centers. To achieve the maximum image fusion level, the coefficient of the transform uses the modified wavelet transform. In the next phase, the fused coefficients of MDWT are given as an input to the convolutional neural network (CNN) classifier. The accuracy of the classifier is enhanced by using the hybrid optimization dynamic (HOD) algorithm. The fused image is classified into malignant or benign by the CNN with the HOD. The HOD is utilized to enhance the classification accuracy and also utilized in optimizing the weights of the CNN.

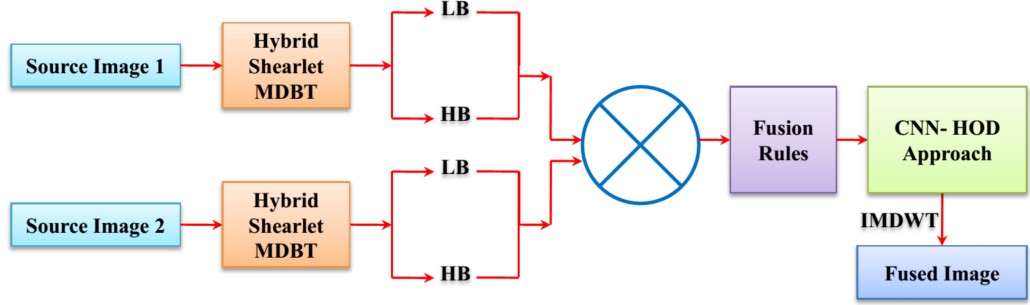

**Figure 1.** Block schematic of the proposed multimodal medical image fusion.



### 3.1. Shearlet Transform

The shearlet transform (ST) occurred as the dominant system through multiscale geometric analysis (MGA) provided with the elegant mathematical form. It is locally multidirectional, well-localized, shift-invariant, multiscale, and ideally sparse. The combined dilation for the affine system is expressed as

$$vj, l(y) = |DETA_M|^{\frac{j}{2}} v\left(s^K J^j y - n\right) : j, k, \in N^2 \tag{1}$$

The anisotropic matrix $A_M$ controls the shearlet scale and the shear matrix $s$ controls the direction. The shift parameters, direction, and scale are denoted by $l$, $k$, and $j$. The invertible matrices are $s$ and $A_M$; it is expressed as

$$A_M = \begin{vmatrix} e & 0 \\ 0 & e^{\frac{j}{2}} \end{vmatrix} \quad \text{and} \quad s = \begin{vmatrix} 1 & S \\ 0 & 1 \end{vmatrix} \tag{2}$$

The below equation expresses the shearlet function and it is computed as

$$\hat{v}^{(0)}(\gamma) = v^{(0)}(\gamma_1, \gamma_2) = \hat{v}_1(\gamma_1)\hat{v}_2\left(\frac{\gamma_2}{\gamma_1}\right) \tag{3}$$

The Fourier transform of $v$ is represented by $\hat{v}$.

$$\sum_{j \geq 0} \left|\hat{v}_1\left(2^{-2j}\eta\right)\right|^2 = 1, \qquad |\eta| \geq 18 \tag{4}$$

For each $j \geq 0$, $v_2$ is

$$\sum_{l=-2^j}^{2^j-1} \left|\hat{v}_2\left(2^j\eta - l\right)\right|^2 = 1, \qquad |\eta| \leq 1 \tag{5}$$

The above Equations (4) and (5) concluded as

$$\sum_{j \geq 0} \sum_{l=-2^j}^{2^j-1} \left|\hat{v}^0(\eta(A_M)_0^{-j}s_0^{-l}\right|^2 = \sum_{j \geq 0} \sum_{l=-2^j}^{2^j-1} \left|\hat{v}_2\left(2^k\frac{\gamma_2}{\gamma_1}\right)\right| \tag{6}$$

From the above equations, the discrete NSST is acquired.

### 3.2. Modified Discrete Wavelet Transform

The input images are decomposed by the modified discrete wavelet transform (DWT). The 1D examination is modified by the multi-resolution examination, which is based on two-dimensional wavelet transform. If $\phi(a)$ and $\eta(a)$ represent the one-dimension scale function and the wavelet function, correspondingly, the subsequent one 2-D scale function as well as three 2-D wavelet functions consist of the foundation of 2-D wavelet transform.

$$\phi(a, b) = \phi(a)\phi(b) \tag{7}$$

$$\left.\begin{array}{l} \eta^U(a) = \phi(a)\eta(b) \\ \eta^G(a) = \eta(a)\phi(b) \\ \eta^E(a) = \eta(a)\phi(b) \end{array}\right\} \tag{8}$$

The L-level decomposition of the image follows $F(a, b)$, and the approximation as well as the three detail transform coefficients are computed.

$$X_M F(a, b) = \langle F(a, b), \phi_M(a, b) \rangle \tag{9}$$

$$E_M^U F(a,b) = \left\langle F(a,b), \eta_M^U(a,b) \right\rangle \tag{10}$$

$$E_M^G F(a,b) = \left\langle F(a,b), \eta_M^G(a,b) \right\rangle \tag{11}$$

$$E_M^E F(a,b) = \left\langle F(a,b), \eta_M^E(a,b) \right\rangle \tag{12}$$

The procedure for using single level wavelet decomposition is as follows:

1. Obtain the original source image as well as the secret image and then obtain the red (R) plane distinctly and establish the single level 2-DDaubechies DWT decomposition on the input source image and the secret image.
2. Let us assume the embedding coefficient is represented as $x$, then the embedding coefficient value is extended from 0 to 1, the coefficient of $x$, and there is a huge rise in robustness and a small rise in transparency.
3. The approximation coefficient is established by utilizing the expression of the horizontal coefficient, diagonal coefficient, and vertical coefficient. The approximation coefficient of the inserted image $= (1 - x)^*$ approximation coefficient of the input image $+x^*$ approximation coefficient of the secret image. In addition, asimilar expression is utilized to compute the diagonal coefficient, horizontal coefficient, and vertical coefficient of the inserted image.
4. Establish the single level 2-D Daubechies inverse DWT decomposition on the computed horizontal, diagonal, approximation, and vertical coefficients to obtain the horizontal, diagonal, approximation, and vertical coefficients of the R plane of the integrated image.
5. The above declared scheme is completed for the blue (B) plane and green (G) plane disjointedly and integrates the blue (B), green (G) and red (R) plane to achieve the integrated image.

### 3.3. CNN-HOD-Based Image Fusion Process

Image fusion represents the procedure of combining the diverse images obtained via the different modalities. During the image fusion process, the transform coefficients obtained from MDWT are given to the CNN-HOD technique to classify the fused images into malignant or benign. The HOD is utilized to enhance the classification accuracy and also to optimize the weights of the CNN. A basic description based on the CNN and HOD optimization algorithm is given in the following section.

#### 3.3.1. Convolutional Neural Network (CNN)

The convolutional neural network (CNN) has had enormous growth in various fields of application for solving problems concerning the classification of images [24]. CNN architecture contains a convolutional layer, pooling layer, and SoftMax layer.

#### Convolutional Layer

The proposed convolutional neural network is comprised of three convolutional layers. The first convolutional layer is utilized to remove numerous low-level features from the input image, namely, edges, corners, and lines. The other two layers of the convolutional network achieve high-level attributes. The attributes of every output map integrates numerous input maps via the convolutions. Normally, the output is specified using the subsequent formula:

$$a_k^m = f\left( \sum_{j \in M_k} a_j^{m-1} * l_{jk}^m * v_k^m \right) \tag{13}$$

where $m$ indicates the $m$th layer, $l_{jk}$ indicates the convolutional kernel, $v_k$ indicates the bias, and $M_k$ indicates the input maps sets. The detailed CNN's implementation utilizes the

sigmoid function, and additive bias is also employed in it. For instance, the unit value at the location ($a$, $b$) in the map of $k$th feature and in the $j$th layer is indicated as

$$y_{jk}^{ab} = sigmoid\left(v_{jk} + \sum_{p=0}^{P_j-1}\sum_{q=0}^{Q_k-1} z_{jk}^{pq} y_{(i-1)}^{(a+p)(b+q)}\right) \tag{14}$$

From the above equation, sigmoid (.) represents the sigmoid function, the feature map bias is indicated as $v_{jk}$, $P_j$ and $Q_k$ represent the kernel height and width, and $z_{jk}^{pq}$ represents the value of the kernel weight at the location ($p$, $q$) associated to the ($j$, $k$) layer. The CNN parameters such as $v_{jk}$ and $z_{jk}^{pq}$ represent the kernel weight.

Pooling Layer

A pooling layer for the sub-sampling layer in a CNN is employed to decrease the variance; it is used to evaluate the highest value over the image of the definite attribute. The pooling layer plays a significant role in the peripheral blood cells classification and recognition. First, the probability $p$ is evaluated for every region $k$ with respect to the below Equation (15).

$$p_j = \frac{\alpha_j}{\sum_{l \in R_k} \alpha_l} \tag{15}$$

The pooling region is represented as $R_k$ in the region k of the feature map and the index of every element is represented as j inside the region. The advantages of this type of implementation are the pooling layer cannot produce the convergence speed of the CNN and also increases the capability of generalization.

SoftMax Layer

A SoftMax layer is employed in the multi-class classification problem. The function of the hypothesis obtains the form:

$$g_\phi(a) = \frac{1}{1 + e^{-\phi^T a}} \tag{16}$$

The major objective of this layer is to train $\phi$ to decreasethe cost function $L(\phi)$.

$$L(\phi) = -\frac{1}{n}\left[\sum_{j=1}^{n}\sum_{k=1}^{m} m\left\{b^{(j)} = k\right\}\log p(b^{(j)} = k|a^{(j)};\phi)\right] \tag{17}$$

The database is trained using $\left\{(a^{(1)}, b^{(1)}), \ldots, (a^{(n)}, b^{(n)})\right\}$, $b^j \in \{1, 2, \ldots, l\}$. The probability of the classification of blood cells $a$ as group $k$ in the softmax layer is:

$$p(b^{(j)} = k|a^{(j)};\phi) = \frac{e^{\phi_k^T a^{(j)}}}{\sum_{m=1}^{l} e^{\phi_m^T a^{(j)}}} \tag{18}$$

The supervised learning approach is utilized for the network training to learn. The internal demonstration replicates the likeness between the training samples. The image attributes are visualized by averaging the patches of the images, which are interrelated by the neurons with a stochastic response in an upper layer. In the classification section, there are two layers, namely, the dense layer and dropout layer. The dense layer is also termed as the fully connected layer, which consists of various neurons or units, whereas the last dense layer consists of several neurons, similar to the number of kinds. After the completion of every dense layer, the activation layer is additionally added. The activation function is employed for the last dense layer output, which is entirely dissimilar to that employed for another dense layer in which the sigmoid or SoftMax function is normally used. A SoftMax layer is used in the multi-class classification process to allocate the probabilities of

the decimal to every kind, and the target kind may have the probability of a high value. The SoftMax of the *j*th output unit is numerically evaluated by the following equation.

$$\hat{b}_j = \frac{e^{a_j}}{\sum_j^M e^{a_j}} \qquad for \quad j = 1, 2, 3, \ldots, M \tag{19}$$

From the above equation, $a_j$ indicates the output of the *j*th dimension. The number of dimensions is represented by $M$, which is equal to the category numbers, and the probability linked with the *j*th category is indicated as $\hat{b}_j$. After the prediction is made, the sample is allocated to the kind which has the probability of a high value as follows:

$$\hat{b}_j = \max_{j \in [1,M]} \hat{b}_j \tag{20}$$

The sigmoid function is employed in the tasks of binary classification. It receives the values of any range of numbers and returns the value that falls in the interval of [0, 1]. This sigmoid function is expressed by the following equation:

$$Sigmoid(x) = \frac{1}{1 + e^{-a}} \tag{21}$$

Dropout layers are the regularization methods implemented only in the network training to forestall it from the problem of overfitting by dropping the subset of the entered neurons and their links momentarily from the last dense layer. The dense layers are normally pursued by the dropout layer, apart from the last dense layer, which generates the kind-particular probabilities. Here, a ResNet model is used as a pre-trained model for the classification of the CNN. The accuracy of the classifier is improved by using the hybrid optimization dynamic algorithm.

### 3.4. Hybrid Optimization Dynamic (HOD) Algorithm

The HOD combines the sailfish optimizer algorithm and the seagull optimization algorithm. The HOD algorithm (see Figure 2) is formed by using both the algorithms. In the sailfish optimizer algorithm, the elitism operation is replaced by the migration operation in the seagull optimization algorithm because the elitism contains the copy of the unaltered fittest solution for the next generation. However, in the seagull optimization algorithm, the migration operation is utilized for finding the fittest solution [25]. In the process of migration, the seagulls are moved as groups. The beginning locations of the seagulls are dissimilar to avoid collisions with each other. In a particular group, the seagulls move in the direction of the optimal seagull.

#### 3.4.1. Sailfish Optimizer Algorithm

The major motivation of the sailfish optimizer algorithm is described in this part. Hence, the proposed algorithm as well as the numerical description is deliberated as follows.

#### Initialization

The sailfish optimizer algorithm represents the population-based metaheuristic algorithm. In this method, the sailfish is assumed as the candidate solutions, and the location of the sailfish is represented as variables in the search space. Therefore, the population over the solution space is arbitrarily created. The selfish search in one, two, three, or hyper dimensional spaces via their variable location vectors. In the e-dimensional search space, the *j*th member at the lth search contains the present location $Sf_{j,l} \in \Re (j = 1, 2, \ldots, n)$. The matrix sailfish is regarded for saving the location of the entire sailfish. Hence, the locations represent the variants of the entire solution through the process of optimization.

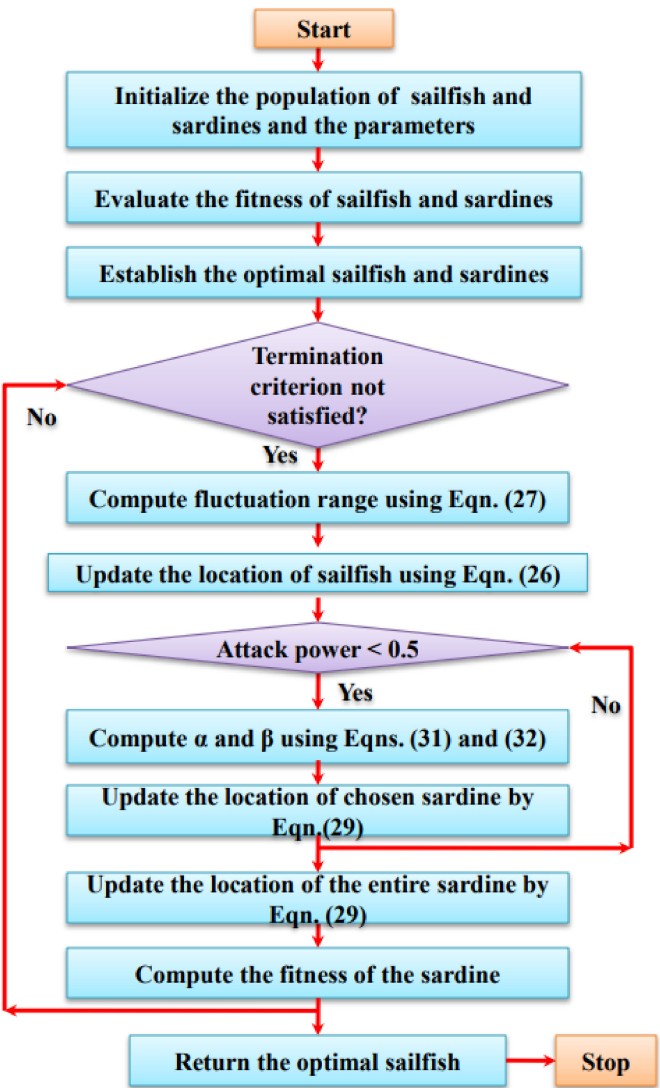

**Figure 2.** Flowchart of HOD algorithm.

$$Sf_{location} = \begin{bmatrix} Sf_{1,1} & Sf_{1,2} & \cdots & Sf_{1,e} \\ Sf_{2,1} & Sf_{2,2} & \cdots & Sf_{2,e} \\ \vdots & \vdots & \vdots & \vdots \\ Sf_{n,1} & Sf_{n,2} & \cdots & Sf_{n,e} \end{bmatrix} \tag{22}$$

where $n$ represents the sailfish numbers, $e$ represents the variable numbers, and $Sf_{j,k}$ represents the value of the $k$th dimension of the $j$th sailfish. Additionally, the fitness of every sailfish is evaluated via the computation of the fitness function as follows:

$$SailfishFitnessValue = F(sailfish) = F(Sf_1, Sf_2, \ldots, Sf_n) \tag{23}$$

Every sailfish is computed using the following matrix that describes the fitness value for the entire solution:

$$Sf_{fitness} = \begin{bmatrix} F(Sf_{1,1} & Sf_{1,2} & \cdots & Sf_{1,e}) \\ F(Sf_{2,1} & Sf_{2,2} & \cdots & Sf_{2,e}) \\ \vdots & \vdots & \vdots & \vdots \\ F(Sf_{n,1} & Sf_{n,2} & \cdots & Sf_{n,e}) \end{bmatrix} = \begin{bmatrix} F_{Sf_1} \\ F_{Sf_1} \\ \vdots \\ F_{Sf_n} \end{bmatrix} \tag{24}$$

where $n$ represents the sailfish numbers, $Sf_{j,k}$ represents the value of the $k$th dimension of the $j$th sailfish, $F$ computes the fitness function, and $Sf_{fitness}$ stores the fitness value, which returns the fitness value for every sailfish. The first row of the $Sf_{location}$ matrix is transmitted to the fitness function, and the output represents the fitness value of the respective sailfish in the $Sf_{fitness}$ matrix.

The sardine group is another important incorporator in the sailfish optimizer algorithm. It is presumed that the sardine group is swimming in the search space. Hence, the sardine location and its fitness values are employed as follows.

$$S_{location} = \begin{bmatrix} S_{1,1} & S_{1,2} & \dots & S_{1,e} \\ S_{2,1} & S_{2,2} & \dots & S_{2,e} \\ \vdots & \vdots & \vdots & \vdots \\ S_{m,1} & S_{m,2} & \dots & S_{m,e} \end{bmatrix} \tag{25}$$

where $m$ represents the sardine numbers, $S_{j,k}$ represents the value of the $k$th dimension of $j$th sardine, and the $S_{location}$ matrix represents the location of the entire sardines.

$$S_{fitness} = \begin{bmatrix} F(S_{1,1} & S_{1,2} & \cdots & S_{1,e}) \\ F(S_{2,1} & S_{2,2} & \cdots & S_{2,e}) \\ \vdots & \vdots & \vdots & \vdots \\ F(S_{m,1} & S_{m,2} & \cdots & S_{m,e}) \end{bmatrix} = \begin{bmatrix} F_{S_1} \\ F_{S_1} \\ \vdots \\ F_{S_m} \end{bmatrix} \tag{26}$$

where $m$ represents the sardine numbers, $S_{j,k}$ describes the value of the $k$th dimension of $j$th sardine, $F$ represents the objective function, and $S_{fitness}$ keeps the fitness value of every sardine. It is prominent that the sailfish as well as the sardines are equivalent factors to compute the solutions. In this method, the sailfish represents the major parameter which is distributed in the search space and sardines cooperate to compute the optimal location in this region. Actually, the sardine is eaten by the sailfish while searching the search space, and the sailfish updates the location to compute the optimal solution achieved up to that point.

Migration

The migration is otherwise called the exploration of seagull modeling. The seagull simulates the seagull group moving towards one location. The seagull swarm movement is scientifically modeled for the method of exploration. There are three rules followed here.

Collision Avoidance

Collision among the neighbors is neglected; the supplementary variable $X$ is utilized for the computation of the location of the new search agent.

$$\vec{Z}_r = X \times \vec{Q}_s(a) \tag{27}$$

where $Z_r$ indicates the location of the search agent that does not collide with the other search agent, $\vec{Q}_s$ indicates the present location of the search agent, $a$ represents the present iteration, and $X$ indicates the movement performance of the search agent in the obtained search space.

$$X = F_z - (a \times (F_z/M_{itr})) \\ where: a = 0, 1, 2, \dots, M_{itr} \tag{28}$$

where $F_z$ is established for controlling the frequency of utilizing variant $X$, which is linearly reduced from $F_z$ to 0. The $F_z$ value is set to 2 for this work.

Moving towards the Direction of the Optimum Seagull

Once the collision among the neighbors is completed, the search agents move in the direction of the optimum neighbor. This activity is completed by satisfying other rules described below.

$$\vec{N}_r = Y \times \left( \vec{Q}_{yr}(a) - \vec{Q}_r(a) \right) \tag{29}$$

where $\vec{N}_r$ indicates the locations of the search agent $\vec{Q}_r$ to the optimum fit search agent $\vec{Q}_{yr}$. The behavior of $Y$ is arbitrative; that is, it is in charge for the appropriate balancing among the exploitation as well as the exploration. The formula for $Y$ is expressed as

$$Y = 2 \times X^2 \times se \tag{30}$$

where $se$ represents the arbitrary number in the interval of [0, 1].

Sustaining Close to the Shortest Distance to the Optimal Search Agent

The search agent updates their location, which is modeled as follows:

$$\vec{E}_r = \left| \vec{Z}_r + \vec{N}_r \right| \tag{31}$$

where $\vec{E}_r$ indicates the distance among the search agent and the optimal fit search agent.

Attack-Interchange Scheme

The sailfish frequently attacks the prey when any of the neighbors are attacked. Sometimes, the sailfish encourages the success rate by the temporarily synchronized attack. The sailfish chases as well as herds its prey. The herding manner of the sailfish changes its location with respect to the position of the other hunters around the prey school, devoid of direct synchronization among them. Consequently, the sailfish update their location inside the sphere around the optimal solution. In the sailfish process, at the $j$th iteration, the novel location of the sailfish $A_{new\_Sf}^{j}$ is updated as follows:

$$A_{new\_Sf}^{j} = A_{elite\_Sf}^{j} - \eta_j \times \left( rand(0,1) \times \left( \frac{X_{new_{Sf}}^{j} + X_{injured_{Sf}}^{j}}{2} \right) - X_{old_{Sf}}^{j} \right) \tag{32}$$

where $A_{elite\_Sf}^{j}$ represents the location of the elite sailfish established up to now, $X_{injured\_S}$ represents the optimum location of the injured sardine established up to now, $X_{old\_S_f}$ represents the present location of the sailfish, $rand(0,1)$ represents the arbitrary number among 0 and 1, and $\eta_j$ represents the coefficient at the $j$th iteration, which is created as follows:

$$\eta_j = 2 \times rand(0,1) \times P_d - P_d \tag{33}$$

where $P_d$ represents the prey density that describes the prey number at every iteration. Due to the reduction in the prey number through the group hunting by sailfish, the factor $P_d$ represents the important factor for the sailfish location update around the prey school. The adaptive expression for this factor is as follows:

$$P_d = 1 - \left( \frac{M_{S_f}}{M_{S_f} + M_S} \right) \tag{34}$$

where $M_{S_f}$ and $M_S$ represents the sailfish numbers and the sardine numbers in every cycle of the algorithm. Additionally, because the primary sardine number is bigger than the sailfish, $M_{S_f}$ is described as $M_S \times P_p$, in which $P_p$ indicates the percentage of the sardine population which establishes the primary sailfish population. With respect to the

average distance among the location of the present optimal sailfish and the present optimal sardine, the location of the sailfish is updated in the iteration course. Using this scheme, the auspicious region of the search space is saved. The sailfish obtain the various places around the school by altering the value of $\eta$. With respect to Equation (33), the variation interval of $\eta$ is in the range of $-1$ and 1, but it is based on the prey number. Otherwise, by reducing $P_d$, the amount of $\eta$ is nearer to $-1$ or 1 according to $rand(0, 1)$ in Equation (33). The factor $\eta$ is leaning towards 1 when $rand(0, 1) > 0.5$; it tends towards $-1$ when $rand(0, 1) > 0.5$, and it is zero for $rand(0, 1) = 0.5$. The fluctuation of $\eta$ and the location of the sailfish is updated from each other and convergence around the prey schools.

Hunting as Well as Catching Prey

At the start of the group hunting, the entire slaughter of the sardines is hardly examined. In most of the cases, sardine scales are eradicated while the sardine bill hits the body of the sardine. This causes the huge sardine number in the schools that contain pronounced injuries on their bodies. At the start of the hunt, the sailfish contain more energy to capture the prey and the sardines are no longer injured and tired. Hence, the sardines maintain a high escape speed and contain a high capability to move. To imitate this procedure, every sardine is gratified for updating the location regarding the present optimal location of the sailfish as well as the power of the attack at every iteration. In the sailfish algorithm, at the $j$th iteration, the new location of the sardine $A^j_{new\_S}$ is expressed as

$$A^j_{new\_S} = s \times \left( A^j_{elite\_Sf} - A^j_{odd_S} + A_P \right) \tag{35}$$

where $A^j_{elite\_Sf}$ represents the optimal location of the elite sailfish established up to now, $A^j_{odd_S}$ represents the present location of the sardine, $s$ represents arbitrary numbers among 0 between 1, and $A_P$ describes the attack power of the sailfish number at every iteration, which is created as follows:

$$A_P = X \times (1 - (2 \times Ite \times \alpha)) \tag{36}$$

where $X$ and $\alpha$ represent the coefficients for reducing the value of the power attack straight from $X$ to 0. To see the consequences of utilizing Equations (35) and (36), they give a few of the probable positions of sardines once slashing of the prey school is finished. Once the sailfish attacks, sardines escape to various places suddenly; then, the sardines update their location to modify the predator and a decrease in the risk is established with respect to $s$ and $A_P$ factors. Actually, the sailfish power attack intensity is reduced, which helps the search agent convergence. By utilizing the parameter of $A_P$, the sardine number updates its location, and the variable numbers are computed as follows:

$$\varepsilon = M_S \times A_P \tag{37}$$

$$\phi = e_j \times A_P \tag{38}$$

where $e_j$ represents the variable number at the $j$th iteration and $M_S$ represents the sardine numbers in every cycle of the algorithm. With respect to the factor of $A_P$, when the sailfish tap intensity is low, then $\varepsilon$ sardines by $\phi$ sardine variables are updated. However, when the sailfish tap intensity is high, the locations of all the sardines are updated. Mostly, $A_P$ and $s$ factors help sailfish optimization to display the arbitrary behavior of the best local stagnation in the entire iterations. The sailfish location is substituted by the recent location of the hunted sardine to increase the chance of hunting new prey, and it is expressed as follows:

$$A^j_{Sf} = A^j_S \; if \; F(S_j) < F(Sf_j) \tag{39}$$

where $A_S^j$ represents the present location of the sardine at the $j$th iteration and $A_{Sf}^j$ represents the present location of the sailfish at the $j$th iteration. In each iteration, the location of every sailfish is updated regarding the elite sailfish and injured sardines. The updating of the location of the sardines is realized using chosen elite sailfish and sardines, which depends upon the sailfish attack power. As the procedure of updating the location of the sardines as well as the sailfish is completed, it is computed by the objective function. The location of the elite sailfish and the injured sardines is updated in every cycle of the algorithm. Then, the hunted sardines are eradicated from the population. These processes are updated iteratively until the termination criterion is gratified.

## 4. Results and Discussions

The proposed approach is verified effectively using 270 pairs of source images. The entire sample of source images was gathered from the Whole Brain Atlas of Harvard Medical School. The examinations were conducted using a set of images that contains CT, MRI, SPECT, MRI, and PET images. The database images are depicted in Figure 3. All the source images contain an identical spatial resolution of 512 × 512 pixels by 256 gray scale levels. The proposed HOD-CNN is computed in MATLAB 2018a with the system requirements being a i7 processor and 8 GB RAM. For the investigation purposes, the database was classified into 75:25 for testing as well as training purposes. Table 2 describes the parameters of the proposed algorithm.

**Table 2.** Parameter settings.

| Techniques | Parameters | Ranges |
|---|---|---|
| Convolutional neural network | Kernel size | 7 × 7 |
| | Learning rate | 0.001 |
| | Batch size | 32 |
| | Optimizer | Adam |
| | Dropout rate | 0.5 |
| Sailfish optimization algorithm | Initial population | 30 |
| | Total iteration | 100 |
| | Fluctuation range | −1 and 1 |
| | Random number | 0 and 1 |
| Seagull optimization algorithm | Population size | 100 |
| | Maximum iterations | 200 |
| | Control parameter | [2, 0] |
| | Frequency control parameter | 2 |

The parameters of various techniques used to tune the proposed method are represented in Table 2.

### 4.1. Performance Measures

The computation of image fusion quality is classified into subjective computation and the objective computations. The performance measures are to choose the appropriate indices to compute the effect of a human visual scheme on image quality perception. The performance of the various approaches is computed based on measures such as edge information retention (QAB/F), average gradient (AG), standard deviation (SD), mutual information (MI), entropy (EN), spatial frequency (SF), and fusion factor (FF). Edge information retention exemplifies the transfer amount of edge detail information in the input images inserted into the fused image. Average gradient is utilized to characterize the image sharpness; if the value of average gradient is large, then the image is clear. Standard deviation describes the reflection of the dispersion degree of the pixel value and also the mean value of the image. If the standard deviation value is greater, then the image quality is better. Mutual information is utilized to compute the information of the fused image present in the utilized image. Entropy exemplifies the amount of information accessible

in the source image as well as the fused image. Spatial frequency represents the entire action of the image in the space domain, and the size is proportional to the consequences of the image fusion. Fusion factor represents the well-identified performance measures that describe the strength of the fusion procedure.

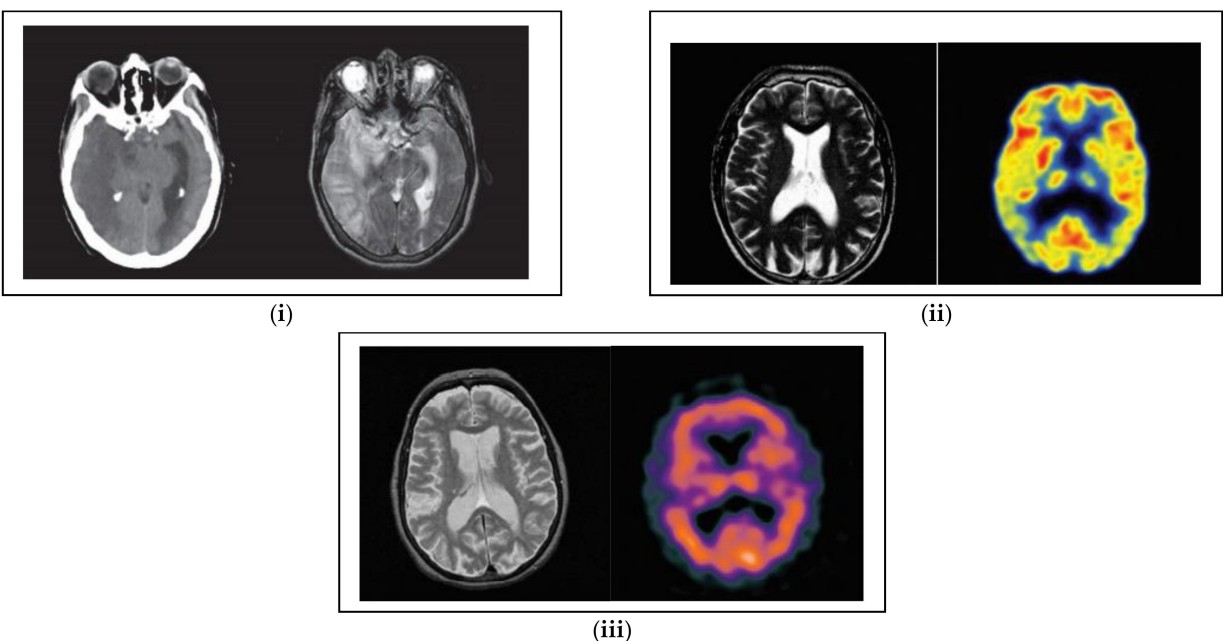

(i)

(ii)

(iii)

**Figure 3.** Database image (**i**) CT and MRI, (**ii**) MRI and PET, and (**iii**) MRI and SPECT.

Figure 4 portrays the graphical analysis to determine the average running time for the proposed approach and various other techniques, namely, NSCT, Kaur et al. (2021);particle swarm optimization (PSO), Shehanaz et al. (2021); convolutional neural network (CNN), Li et al. (2021); and adolescent identity search (AIS) algorithm, Jose et al. (2021). The graph is plotted for the running time and various approaches. From the experimentation, the evaluation results revealed that the proposed approach attains a minimum average running time of about 0.53 s compared with other approaches.

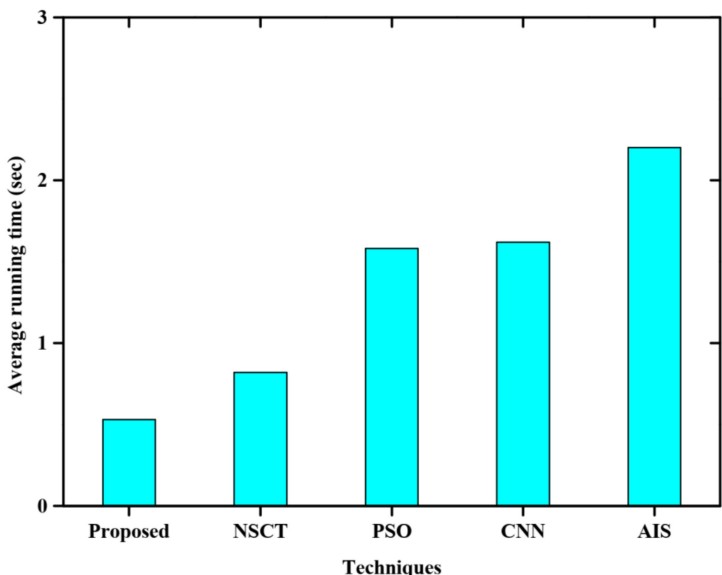

**Figure 4.** Average running time analysis.

### 4.2. Quantitative Analysis

The fused images for the three dataset images are shown in Figures 5–7. In this part, the proposed approach is compared with the existing approaches such as NSST-PAPCNN [20], EMBO-DST [12], DNN-DGSA [14], and PCNN-NSST [15] by regarding the parameters such as edge information retention (QAB/F), average gradient (AG), standard deviation (SD), mutual information (MI), entropy (EN), spatial frequency (SF), and fusion factor (FF). Figure 4 describes the fusion results of CT and MRI. Figure 5 describes the fusion results of CT and PET images. Figures 6 and 7 describe the fusion results of the CT and PET images as well as the CT and SPECT images. From the comparative analysis, the results reveal that the proposed approach attains better performances than with other approaches. Tables 3–5 describe the objective computation of various methods on medical image fusion for CT and MRI, CT and PET images, and CT and SPECT images. The results (Table 6) show that the proposed approach has better results when compared with other approaches.

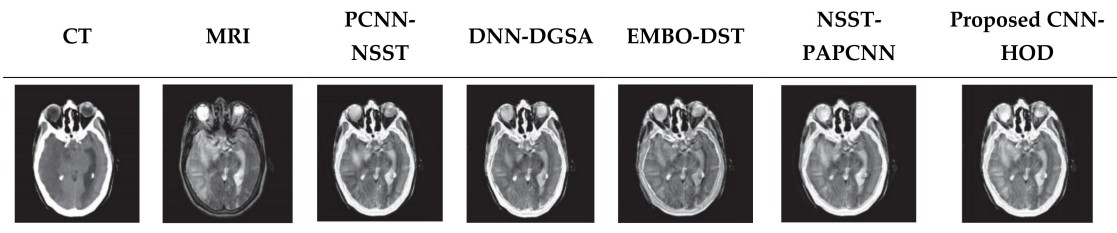

| CT | MRI | PCNN-NSST | DNN-DGSA | EMBO-DST | NSST-PAPCNN | Proposed CNN-HOD |

**Figure 5.** Fusion results of CT and MRI.

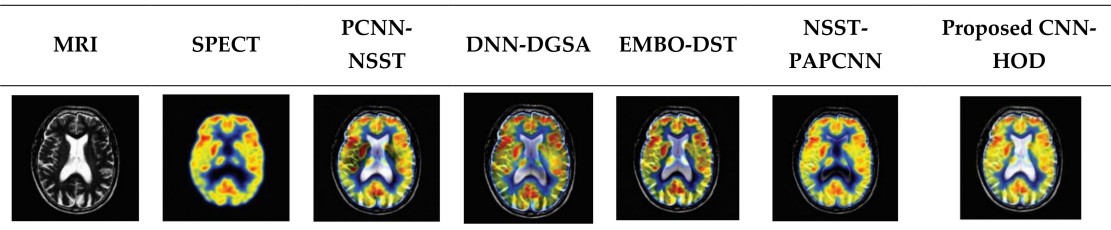

| MRI | SPECT | PCNN-NSST | DNN-DGSA | EMBO-DST | NSST-PAPCNN | Proposed CNN-HOD |

**Figure 6.** Fusion results of CT and PET images.

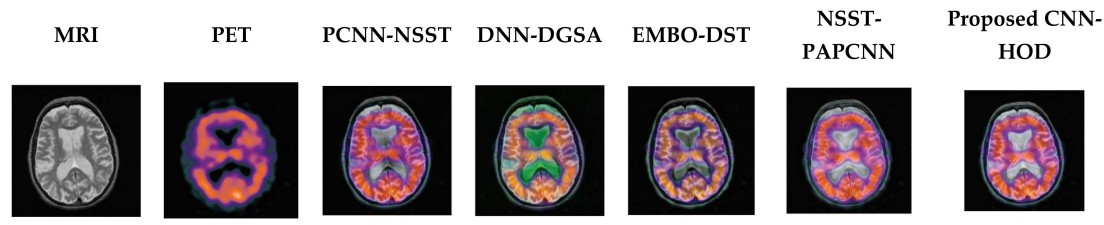

| MRI | PET | PCNN-NSST | DNN-DGSA | EMBO-DST | NSST-PAPCNN | Proposed CNN-HOD |

**Figure 7.** Fusion results of CT and SPECT images.

**Table 3.** Objective computation of various methods on medical image fusion for CT and MRI.

| Measures | PCNN-NSST | DNN-DGSA | EMBO-DST | NSST-PAPCNN | Proposed |
|----------|-----------|----------|----------|-------------|----------|
| QAB/F | 0.2082 | 0.2284 | 0.2653 | 0.3645 | 0.4672 |
| AG | 6.3128 | 6.6754 | 6.9816 | 7.6542 | 7.8914 |
| SD | 47.2761 | 48.1692 | 50.7616 | 51.6723 | 54.6870 |
| MI | 2.6892 | 2.7654 | 2.8974 | 3.1678 | 3.4152 |
| EN | 4.4264 | 4.5298 | 4.6784 | 4.8532 | 4.9952 |
| SF | 19.9757 | 20.7865 | 21.6738 | 23.8761 | 24.7622 |
| FF | 5.9824 | 6.0935 | 6.1382 | 6.5665 | 8.3281 |

**Table 4.** Objective computation of various methods on medical image fusion for CT and PET images.

| Measures | PCNN-NSST | DNN-DGSA | EMBO-DST | NSST-PAPCNN | Proposed |
|---|---|---|---|---|---|
| QAB/F | 0.3186 | 0.3484 | 0.3941 | 0.4457 | 0.5392 |
| AG | 6.1326 | 6.8331 | 7.9814 | 8.0642 | 8.4631 |
| SD | 46.1488 | 49.9126 | 51.8674 | 53.5648 | 55.4872 |
| MI | 2.9827 | 3.2673 | 3.8915 | 4.2186 | 4.6524 |
| EN | 4.5148 | 4.7528 | 4.9921 | 5.0885 | 5.1872 |
| SF | 22.8907 | 25.8733 | 28.9154 | 30.7372 | 32.8245 |
| FF | 6.1429 | 6.4634 | 6.8736 | 7.1984 | 7.3562 |

**Table 5.** Objective computation of various methods on medical image fusion for CT and SPECT images.

| Measures | PCNN-NSST | DNN-DGSA | EMBO-DST | NSST-PAPCNN | Proposed |
|---|---|---|---|---|---|
| QAB/F | 0.4177 | 0.4575 | 0.5884 | 0.6429 | 0.7362 |
| AG | 7.7648 | 7.9731 | 8.1583 | 8.6758 | 8.8763 |
| SD | 50.8734 | 52.7615 | 54.8324 | 56.7522 | 58.6421 |
| MI | 4.2541 | 4.4659 | 4.9826 | 5.0942 | 5.1644 |
| EN | 4.4328 | 4.6715 | 4.8259 | 5.2781 | 5.4638 |
| SF | 27.5714 | 29.8765 | 30.9816 | 32.7625 | 34.8712 |
| FF | 6.9876 | 7.6978 | 7.8573 | 8.2538 | 8.7642 |

**Table 6.** Evaluation of fusion results.

| Methods | Average Gradient | Fusion Factor | Standard Deviation |
|---|---|---|---|
| DWT | 6.5342 | 7.0346 | 50.4563 |
| Shearlet | 7.6859 | 7.2785 | 53.098 |
| Contourlet | 7.8219 | 7.8654 | 55.5231 |
| MDWT | 8.2731 | 8.0457 | 57.4563 |
| Hybrid MDWT-Shearlet | 8.9142 | 8.8012 | 59.7314 |

## 5. Conclusions

In this paper, fused multimodality medical image classification is proposed depending upon a CNN with HOD. The major role of this method is optimal fusion using the hybrid optimization dynamic algorithm. Initially, multimodal medical images are obtained for the fusion process using modified discrete wavelet transform (MDWT). The fused image is classified into malignant or benign using a convolutional neural network (CNN). The HOD is utilized for enhancing the classification accuracy of the CNN algorithm. The HOD contains the sailfish optimizer algorithm and seagull optimization algorithm. The seagull optimizer algorithm replaces the migration operation to obtain the optimal location. Experimental analysis is carried out and compared with the other approaches with respect to performance measures such as edge information retention (QAB/F), average gradient (AG), standard deviation (SD), mutual information (MI), entropy (EN), spatial frequency (SF), and fusion factor (FF). The proposed approach is compared with other approaches on the databases and it is revealed that the proposed approach produced improved results. The experimental results show that the proposed approach performs better than other approaches and offers high quality fused images for an accurate diagnosis. In the future, the proposed approach has to be implemented in real-time applications and employed for other kinds of multimodality medical image fusion such asmulti-focus image fusion and infrared visible.

**Author Contributions:** Conceptualization, M.M.A.; Data curation, A.M.A.; Methodology, M.M.A.; Software, A.M.A.; Supervision, A.M.A.; Writing—original draft, M.M.A.; Writing—review & editing, M.M.A. All authors have read and agreed to the published version of the manuscript.

**Funding:** This research received no external funding.

**Institutional Review Board Statement:** The study was conducted according to the guidelines of the Declaration of Helsinki, and approved by the Institutional Review Board.

**Data Availability Statement:** Not applicable.

**Conflicts of Interest:** The authors declare no conflict of interest.

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
