# Peer review of "Artificial Intelligence-Based Multimodal Medical Image Fusion Using Hybrid S2 Optimal CNN"

_electronics, doi:10.3390/electronics11142124_

Round 1

Reviewer 1 Report

This paper proposed a novel modified DWT and CNN based fusion methods. The proposed method obtained better performance according to the limited experiments.

However, I still have some concerns.

  1. Abbreviations("HOD") should be explained before they are used.
  2. What's the inputs and the outputs of CNN-HOD model? In Section 3, it says the fused images are fed into CNN-HOD. But from Figure 1, the inputs are the fused MDWT coefficients.
  3. The fusion rules are not well introduced.
  4. There are many deep learning-based medical image fusion methods, I suggest the authors could do more experiments with these novel methods (published in 2021).
  5. Some ablation studies should be conducted, such why use MDWT (should give some experimental data).

Author Response

Comments and Suggestions for Authors 1

This paper proposed a novel modified DWT and CNN based fusion methods. The proposed method obtained better performance according to the limited experiments.

However, I still have some concerns.

Comments: Abbreviations("HOD") should be explained before they are used.

Authors Response: Thank you for your comment. As per your suggestion, we have mentioned the abbreviation of the HOD algorithm at the initial section of the manuscript. (Refer abstract section)

Comments: What's the inputs and the outputs of CNN-HOD model? In Section 3, it says the fused images are fed into CNN-HOD. But from Figure 1, the inputs are the fused MDWT coefficients.

Authors Response: Thank you for your comment. As per your suggestion, we have solved the above mentioned comments in the revised manuscript. The transform coefficients of MDWT are provided as an input and the fused image is obtained as an output using CNN-HOD algorithm. (Refer section 3)

Comments: The fusion rules are not well introduced.

Authors Response: Thank you for your comment. As per your suggestion, we have provided the detailed regarding the image fusion process in the revised manuscript. Image fusion represents the procedure of combining the diverse images obtained by the different modalities. During image fusion process, the transform coefficients obtained from MDWT is given to the CNN-HOD technique for fusing images. The HOD is utilized to enhance the classification accuracy and also utilized to optimize the weights of the CNN. (Refer section 3.2).

Comments: There are many deep learning-based medical image fusion methods, I suggest the authors could do more experiments with these novel methods (published in 2021).

Authors Response: Thank you for your comment. As per your suggestion, we have made experimental analysis for various novel techniques in the revised manuscript. Fig.4 portrays the graphical analysis to determine the average running time for the proposed approach and various other techniques namely the NSCT, Kaur et al. (2021); Particle swarm optimization (PSO), Shehanaz et al. (2021); convolutional neural network (CNN), Li et al. (2021); as well as adolecent identity search (AIS) algorithm, Jose et al. (2021). The graph is plotted for the running time and various approaches. From the experimentation, the evaluation results revealed that the proposed approach attains minimum average running time of about 0.53 seconds than other approaches. (Refer fig.4)

Comments: Some ablation studies should be conducted, such why use MDWT (should give some experimental data).

Authors Response: Thank you for your comment. When compared with other wavelet transforms, the key advantage of MDWT technique is that, it has over Fourier transforms is temporal resolution as it captures both frequency and location information.

Reviewer 2 Report

The manuscript by Almasri et al. described a multimodal medical image fusion using Convolutional neural network (CNN) algorithm and Hybrid optimization dynamic (HOD) algorithm. The authors then compared their fusion method with the other approaches through several parameters, such as edge information retention, average gradient, standard deviation, mutual information, entropy, spatial frequency and fusion factor. The experimental results seem to show that their proposed approach performs better than other approaches by providing images of better quality.

My primary interests in reading the manuscript are with aspects of how distinguishable the fused images compared to one made by other methods and how much the CNN-HOD method improves the fused image quality. The answer seems to be presented in Tables 1, 2, 3, however, no discussion was made on these obtained parameters, including QAB/F, AG, SD, MI, EN, SF and FF. For example, on the merged CT and MRI images, is the change of FF from 6.5665 (by the approach of NSST-PAPCNN) to 6.8762 (by CNN-HOD) considered a significant improvement? Which parameter can only be best obtained by CNN-HOD method? Because the discussion on Quantitative Analysis (Section 4.2) is weak, the last a few sentences in the Abstract section becomes extremely vague. E.g., “the proposed approach performs better with respect to the measures than other approaches and offers high quality fused images for an accurate diagnosis”: one would ask “how much better” and “how much higher quality”? I do not see any conclusive numbers on these quantitative improvements, despite of the quantified parameters in Tables 1, 2, 3. On the other hand, to apply this approach method in real medical imaging and analysis a large test on numerous images is required. This throws out another question, is the improvement on image quality (based on the quantified parameters) reproducible and valid for other imaging data?

I also think the writing need very careful polishing to improve the reading and scientific flow. Please check the minor comments below.

- Page 1, Line 22, “show”;

- Page 1, Line 42, a review of the biomedical imaging using different modalities should be included in the Introduction part for content completeness and expanding broader readership, e.g., Acc. Chem. Res. 2011, 44, 10, 1050–1060.

- Page 2, Line 62, “based”

- Page 2, Line 89, delete “).”, and try this sentence again: “The major drawback was low image quality and performance was not consistent so low efficiency”

- Page 3, Line 99, the author is mixed using “MRI” with “MR” everywhere. The correct term is MRI or MR imaging.

- Page 3, Line 101, delete “using this method”

- Page 3, Line 107, change “difficult to implement” to “the implementation”, this occurs many times, .e.g., Page 3, Lines 118, 121, etc.

- Page 3, Line 108, change “Tanet al.” to “Tan et al.”. The author should use the correct form of “surname + et al.” NOT “initial/first name + et al.”, and this occurs everywhere, e.g., same page Lines 122, 126, Table 1, etc.

- Page 4, Line 144, “to optimize”

- Figure 1, redraw the arrow line without extra ends

- Page 6, Line 186, “equipment”

Author Response

Comments and Suggestions for Authors 2

The manuscript by Almasri et al. described a multimodal medical image fusion using Convolutional neural network (CNN) algorithm and Hybrid optimization dynamic (HOD) algorithm. The authors then compared their fusion method with the other approaches through several parameters, such as edge information retention, average gradient, standard deviation, mutual information, entropy, spatial frequency and fusion factor. The experimental results seem to show that their proposed approach performs better than other approaches by providing images of better quality.

My primary interests in reading the manuscript are with aspects of how distinguishable the fused images compared to one made by other methods and how much the CNN-HOD method improves the fused image quality. The answer seems to be presented in Tables 1, 2, 3, however, no discussion was made on these obtained parameters, including QAB/F, AG, SD, MI, EN, SF and FF. For example, on the merged CT and MRI images, is the change of FF from 6.5665 (by the approach of NSST-PAPCNN) to 6.8762 (by CNN-HOD) considered a significant improvement? Which parameter can only be best obtained by CNN-HOD method? Because the discussion on Quantitative Analysis (Section 4.2) is weak, the last a few sentences in the Abstract section becomes extremely vague. E.g., “the proposed approach performs better with respect to the measures than other approaches and offers high quality fused images for an accurate diagnosis”: one would ask “how much better” and “how much higher quality”? I do not see any conclusive numbers on these quantitative improvements, despite of the quantified parameters in Tables 1, 2, 3. On the other hand, to apply this approach method in real medical imaging and analysis a large test on numerous images is required. This throws out another question, is the improvement on image quality (based on the quantified parameters) reproducible and valid for other imaging data?

Authors Response: Thank you for your comment. As per your suggestion we have solved the above mentioned comment in the revised manuscript. Edge information retention exemplifies the transfer amount of edge detail information in the input images inserted into the fused image; Average gradient is utilized to characterize the image sharpness, if the value of average gradient is large, then the image is clear; Standard deviation describes it reflects the dispersion degree of the pixel value and also the mean value of the image. If the standard deviation value is greater, then the image quality is better; Mutual information is utilized to compute the information of the fused image present in the utilized image; Entropy exemplifies the number of information accessible in the source image as well as the fused image; Spatial frequency represents the entire action of the image in the space domain and the size is proportional to the consequences of the image fusion; Fusion factor represents the well-identified performance measures that describes the strength of the fusion procedure.

Comments: I also think the writing need very careful polishing to improve the reading and scientific flow. Please check the minor comments below.

- Page 1, Line 22, “show”;

- Page 1, Line 42, a review of the biomedical imaging using different modalities should be included in the Introduction part for content completeness and expanding broader readership, e.g., Acc. Chem. Res. 2011, 44, 10, 1050–1060.

- Page 2, Line 62, “based”

- Page 2, Line 89, delete “).”, and try this sentence again: “The major drawback was low image quality and performance was not consistent so low efficiency”

- Page 3, Line 99, the author is mixed using “MRI” with “MR” everywhere. The correct term is MRI or MR imaging.

- Page 3, Line 101, delete “using this method”

- Page 3, Line 107, change “difficult to implement” to “the implementation”, this occurs many times, .e.g., Page 3, Lines 118, 121, etc.

- Page 3, Line 108, change “Tanet al.” to “Tan et al.”. The author should use the correct form of “surname + et al.” NOT “initial/first name + et al.”, and this occurs everywhere, e.g., same page Lines 122, 126, Table 1, etc.

- Page 4, Line 144, “to optimize”

- Figure 1, redraw the arrow line without extra ends

- Page 6, Line 186, “equipment”

Authors Response: Thank you for your comment. As per your suggestion, we have solved the above mentioned comment and highlighted in red color in the revised manuscript. (Refer manuscript)

Reviewer 3 Report

The authors studied Medical image fusion based on deep learning convolutional neural network and hybrid optimization dynamic algorithm. The idea of proposing a multimodality classification for a medical image is not new as lots of papers have already been published by using different methods; the only novelty is the use of a hybrid optimization dynamic algorithm as an optimizer function. The presentation is not organized logically. Also, the quality of written English is poor, I would suggest that the authors have their manuscripts checked by an English language native speaker.

Some suggestions for further improvement:

The summary abstract is not concise. I do recommend the re-writing of the abstract and grammar check through the document.

The Introduction section is well-written, and I propose to schematize all the discussion in several paragraphs (to facilitate the reading): (1) motivations, (2) the overall approach, (3) main contributions.

However, the main challenges to the field and the needs and benefits of this study are missed in the introduction.

A general discussion of the limitations and expectations of the proposed model should be inserted.

The dataset with only 88 pairs of images is too small for training CNN. I suggest authors test their model with another dataset and add their results in the “Results and discussions” section.

The 'CNN supervised' approach used in the model is not explained well. There is a brief mention that it can be intuitively deduced in a referenced work, but at least a basic description should also be present in this work.

The authors didn't have performed a sufficient analysis of the uncertainties in their results, for example, the numbers of significant figures and tables are not justified. The need for statistical analysis is required to make statements of the significance of differences in results. The authors have to explicitly study the uncertainties of the experimental variables that represent the key to the measurements.

Please re-check and change all the abbreviations inside the manuscript. For instance: hybrid optimization dynamic (HOD) is mentioned a lot in the full version, inside discussion and conclusion sections which need to be abbreviated.

Please add the number of parameters of your model in the “Results and discussions” section.

I assume that the authors used a pre-trained model for the CNN classification. Please mentioned which pre-trained model you used? (DenseNet, ResNet, …)

Authors need to explain all figures that exist in the manuscripts. Please explain Figure 4, Figure 5, and Figure 6 in text.

Line 203 – 204: “kernel weight and it is generally trained by employing unsupervised methods” what does this sentence means? The authors mentioned that their model is supervised learning, therefore the weights will be updated through backpropagation, not unsupervised methods!

Line 449: Please add a space between the dot and “The” word.

Finally, given all the above. I suggest to MAJOR REVISION the paper.

Author Response

Comments and Suggestions for Authors 3

The authors studied Medical image fusion based on deep learning convolutional neural network and hybrid optimization dynamic algorithm. The idea of proposing a multimodality classification for a medical image is not new as lots of papers have already been published by using different methods; the only novelty is the use of a hybrid optimization dynamic algorithm as an optimizer function. The presentation is not organized logically. Also, the quality of written English is poor, I would suggest that the authors have their manuscripts checked by an English language native speaker.

Some suggestions for further improvement:

Comments: The summary abstract is not concise. I do recommend the re-writing of the abstract and grammar check through the document.

Authors Response: Thank you for your comment. As per your suggestion, we have made corrections in the abstract and solved the above mentioned comment in the revised manuscript. (Refer abstract)

Comments: The Introduction section is well-written, and I propose to schematize all the discussion in several paragraphs (to facilitate the reading): (1) motivations, (2) the overall approach, (3) main contributions.

Authors Response: Thank you for your comment. As per your suggestion, we have schematized all the discussion in several paragraphs in the introduction section of the revised manuscript. (Refer section 1)

Comments: However, the main challenges to the field and the needs and benefits of this study are missed in the introduction.

Authors Response: Thank you for your comment. As per your suggestion, we have mentioned the major challenges and the benefits of the study in the revised manuscript. (Refer section 1)

Comments: A general discussion of the limitations and expectations of the proposed model should be inserted.

Authors Response: Thank you for your comment. As per your suggestion, we have discussed the limitations and expectations of the proposed model in the revised manuscript. (Refer section 5)

Comments: The dataset with only 88 pairs of images is too small for training CNN. I suggest authors test their model with another dataset and add their results in the “Results and discussions” section.

Authors Response: Thank you for your comment. The proposed approach is verified effectively using 270 pairs of the source images. The entire source images are gathered from the Whole Brain Atlas of Harvard medical school. The examinations were conducted using the set of the images that contains CT and MRI and SPECT, MRI and PET images. The database images are depicted in Fig 3. The entire source images contain the identical spatial resolution of 512 x 512 pixels by 256 gray scale levels. (Refer section 4)

Comments: The 'CNN supervised' approach used in the model is not explained well. There is a brief mention that it can be intuitively deduced in a referenced work, but at least a basic description should also be present in this work.

Authors Response: Thank you for your comment. As per your suggestion, we have explained the CNN technique in the revised manuscript. (Refer section 3.2.1)

Comments: The authors didn't have performed a sufficient analysis of the uncertainties in their results, for example, the numbers of significant figures and tables are not justified. The need for statistical analysis is required to make statements of the significance of differences in results. The authors have to explicitly study the uncertainties of the experimental variables that represent the key to the measurements.

Authors Response: Thank you for your comment. As per your suggestion, we have performed a sufficient analysis of the uncertainties in the result section and we have provided the experimental analysis for the proposed approach and compared with various other techniques in the revised manuscript. (Refer result section)

Comments: Please re-check and change all the abbreviations inside the manuscript. For instance: hybrid optimization dynamic (HOD) is mentioned a lot in the full version, inside discussion and conclusion sections which need to be abbreviated.

Authors Response: Thank you for your comment. As per your suggestion, we have rechecked and changed all the abbreviations inside the manuscript. (Refer manuscript)

Comments: Please add the number of parameters of your model in the “Results and discussions” section.

Authors Response: Thank you for your comment. As per your suggestion, we have added the parameter settings of the proposed model in the revised manuscript. (Refer table 1)

Comments: I assume that the authors used a pre-trained model for the CNN classification. Please mentioned which pre-trained model you used? (DenseNet, ResNet, …)

Authors Response: Thank you for your comment. As per your suggestion, we have mentioned the type of classifier in the revised manuscript. In this paper, a ResNet model is used as a pre trained model for the classification of CNN.(Refer section 3.2)

Comments: Authors need to explain all figures that exist in the manuscripts. Please explain Figure 4, Figure 5, and Figure 6 in text.

Authors Response: Thank you for your comment. As per your suggestion, we have provided the explanations  for all the figures in the revised manuscript. (Refer figures)

Comments: Line 203 – 204: “kernel weight and it is generally trained by employing unsupervised methods” what does this sentence means? The authors mentioned that their model is supervised learning, therefore the weights will be updated through back propagation, not unsupervised methods!

Authors Response: Thank you for your comment. As per your suggestion, we have solved the above mentioned comment in the revised manuscript. (Refer section 3.2.1)

Comments: Line 449: Please add a space between the dot and “The” word.

Authors Response: Thank you for your comment. As per your suggestion, we have solved the above mentioned comment in the revised manuscript.

Reviewer 4 Report

This paper proposes a novel technique for fusing different modalities of medical images (particularly CT & {MR, PET, SPECT} images) using a multistage workflow. Images are initially combined using a modified discrete wavelet transform followed by a customized CNN architecture. The utility of the proposed algorithm is validated by computing quantitative measures of fused image quality for what is assumed to be a  publically-available dataset (“Whole Brain Atlas of Harvard Medical School,” which is not explicitly referenced within the text). Results are benchmarked versus many other recent results in this space, with the proposed algorithm demonstrating superior performance in each metric considered.

In my opinion, the primary weaknesses of the paper are as follows – 1) as presented, the motivation of the paper seems limited and the contributions are nearly non-existent and 2) the methods employed (both for model development and verification) are poorly described and no link to the utilized software is provided.    

Regarding 1), the best evidence for this is gained by viewing the stated contributions by the authors beginning on line 80 (ie: To decompose the images into low and high-frequency sub bands using modified discrete wavelet transform (MDWT), etc.). As noted immediately after in the related work section by the authors themselves, this application is in itself not novel. To improve the quality of the paper, the authors must clearly delineate their contributions (perhaps it is in the superior performance methods themselves). Regarding motivation, there is little clear commentary on why the particular architecture was chosen, and should be expected to yield superior results to other comparative architectures. This detail must be added in my opinion to provide value to the literature.

In addition, the quality of the description of both the authors’ network architecture, along with the process used for validation needs to be considerably improved. An anecdote of the need for additional detail is emphasized in Fig 1, the key block diagram of the paper, where terms such as “LB” and “HB” are used without any subsequent reference in the paper. Beyond describing their own work in an improved manner (including the potential streamlining/limitation of fundamental details), details regarding how validation was performed must also be improved (ie: add a link to the dataset, mention explicitly why this dataset was chosen (was it used by other benchmark work), state how the benchmark models were implemented, etc.).

Beyond these two limitations, the paper would also benefit from considerable editing. This is evidenced by a collection of small mistakes throughout the paper (ie: 2nd line of the abstract, in which “themedical” is combined into one word, line 52 where “techniquesaim” is a single word, line 144 where “tooptimize” is a single word, etc.). Beyond this basic proof-reading, the paper would also benefit from improving the quality of writing throughout the manuscript.

Finally, the paper could also be improved by enhancing the quality of provided references. For example, in the case of citing the broad utility of medical image fusion within the introduction, only a single reference to a recent paper. Citing instead to one of the many recent reviews of the state of the art on this topic would be a much better approach in my opinion. In addition, increasing the number of references throughout the introduction would also be valuable (for example, citing a well-referenced work which utilizes the taxonomy proposed on line 49).

Please find some other issues that must be clarified or addressed listed below –

  • The HOD acronym is first utilized in line 73 but not defined until later in the paper.
  • While the level of mathematical detail describing the processing steps beginning on line 154 is appreciated, the clarity of this section must be improved.
  • Within this aforementioned section, there is reference to RGB planes. Does this imply that the input image type is restricted to traditional 3 channel images?
  • The phrase “secret image” is introduced in line 158. Perhaps I am simply unfamiliar with the relevant literature in which this phrase would be known without introduction. However, I think it may merit providing an introduction herein for purposes of completeness.
  • While the attempt to benchmark the proposed approach versus prior work (ie: PCNN-NSST, etc.) is greatly appreciated, it would be nice if the authors clarified how the corresponding algorithms were implemented (ie: open source implementation, replication by the authors themselves, etc.).

Author Response

Comments and Suggestions for Authors 4

This paper proposes a novel technique for fusing different modalities of medical images (particularly CT & {MR, PET, SPECT} images) using a multistage workflow. Images are initially combined using a modified discrete wavelet transform followed by a customized CNN architecture. The utility of the proposed algorithm is validated by computing quantitative measures of fused image quality for what is assumed to be a publically-available dataset (“Whole Brain Atlas of Harvard Medical School,” which is not explicitly referenced within the text). Results are benchmarked versus many other recent results in this space, with the proposed algorithm demonstrating superior performance in each metric considered. In my opinion, the primary weaknesses of the paper are as follows – 1) as presented, the motivation of the paper seems limited and the contributions are nearly non-existent and 2) the methods employed (both for model development and verification) are poorly described and no link to the utilized software is provided.    

Comments: authors beginning on line 80 (ie: To decompose the images into low and high-frequency sub bands using modified discrete wavelet transform (MDWT), etc.). As noted immediately after in the related work section by the authors themselves, this application is in itself not novel. To improve the quality of the paper, the authors must clearly delineate their contributions (perhaps it is in the superior performance methods themselves). Regarding motivation, there is little clear commentary on why the particular architecture was chosen, and should be expected to yield superior results to other comparative architectures. This detail must be added in my opinion to provide value to the literature.

Authors Response: Thank you for your comment. As per your suggestion, we have solved the above mentioned comment in the revised manuscript. Also, we have changed the contribution of the paper that is mentioned below. A modified discrete wavelet transform (MDWT) is utilized to decompose the images into low and high-frequency sub bands. The fused images are classified as malignant or benign using the proposed convolutional neural network based hybrid optimization dynamic algorithm (CNN-HOD).To compare the proposed approach with various other image fusion based techniques to evaluate the performance of the system. (Refer contributions in section 1)

Comments: In addition, the quality of the description of both the authors’ network architecture, along with the process used for validation needs to be considerably improved. An anecdote of the need for additional detail is emphasized in Fig 1, the key block diagram of the paper, where terms such as “LB” and “HB” are used without any subsequent reference in the paper. Beyond describing their own work in an improved manner (including the potential streamlining/limitation of fundamental details), details regarding how validation was performed must also be improved (ie: add a link to the dataset, mention explicitly why this dataset was chosen (was it used by other benchmark work), state how the benchmark models were implemented, etc.).

Authors Response: Thank you for your comment. This paper proposed a novel hybrid dynamic optimization algorithm. The HOD combines the sailfish optimizer algorithm and seagull optimization algorithm. The HOD algorithm is formed by using both the algorithms thereby enhancing the convergence probability and minimizing the local optimal value. Inthe sailfish optimizer algorithm, the elitism operation is replaced by the migration operation in the seagull optimization algorithm because the elitism contains the copy of the unaltered fittest solution for the next generation. But in the seagull optimization algorithm, the migration operation is utilized for finding the fittest solution. (Refer section 3)

Comments: Beyond these two limitations, the paper would also benefit from considerable editing. This is evidenced by a collection of small mistakes throughout the paper (ie: 2nd line of the abstract, in which “themedical” is combined into one word, line 52 where “techniquesaim” is a single word, line 144 where “tooptimize” is a single word, etc.). Beyond this basic proof-reading, the paper would also benefit from improving the quality of writing throughout the manuscript.

Authors Response: Thank you for your comment. As per your suggestion, we have corrected the above mentioned errors and solved the grammatical mistakes in the revised manuscript. (Refer abstract section)

Comments: Finally, the paper could also be improved by enhancing the quality of provided references. For example, in the case of citing the broad utility of medical image fusion within the introduction, only a single reference to a recent paper. Citing instead to one of the many recent reviews of the state of the art on this topic would be a much better approach in my opinion. In addition, increasing the number of references throughout the introduction would also be valuable (for example, citing a well-referenced work which utilizes the taxonomy proposed on line 49).

Authors Response: Thank you for your comment. As per your suggestion, we have added recent papers in the literature review section of the revised manuscript. (Refer section 2)

Please find some other issues that must be clarified or addressed listed below –

Comments: The HOD acronym is first utilized in line 73 but not defined until later in the paper.

Authors Response: Thank you for your comment. As per your suggestion, we have mentioned the abbreviation of the HOD algorithm at the initial section of the manuscript. (Refer abstract section)

Comments: While the level of mathematical detail describing the processing steps beginning on line 154 is appreciated, the clarity of this section must be improved.

Authors Response: Thank you for your comment. As per your suggestion, we have improved the pre-processing steps in the revised manuscript. (Refer section 3)

Comments: Within this aforementioned section, there is reference to RGB planes. Does this imply that the input image type is restricted to traditional 3 channel images?

Authors Response: Thank you for your comment. Yes, the scheme is completed for the Blue (B) plane and Green (G) plane disjointedly and integrates Blue (B), Green (G) and Red (R) plane to achieve the integrated image.

Comments: The phrase “secret image” is introduced in line 158. Perhaps I am simply unfamiliar with the relevant literature in which this phrase would be known without introduction. However, I think it may merit providing an introduction herein for purposes of completeness.

Authors Response: Thank you for your comment. Secret image is that confidentiality needs to be maintained, and also to authenticate the distributor who distributes that secret image to multiple users

Comments: While the attempt to benchmark the proposed approach versus prior work (ie: PCNN-NSST, etc.) is greatly appreciated, it would be nice if the authors clarified how the corresponding algorithms were implemented (ie: open source implementation, replication by the authors themselves, etc.).

Authors Response: Thank you for your comment. The corresponding algorithms are replication by the author themselves.

Round 2

Reviewer 1 Report

Many thanks to the authors for the efforts on the revised version.
I still have one main concerns.

For the ablation study, what my suggestion is that the authors should replace the proposed MDWT module with other transform operations, such as contourlet, shearlet etc. I suggest the authors could give some experimental data about this ablation study.

Author Response

Response: We first of all, thank the reviewer for the positive insights in the manuscript. As a response to the above comment, the proposed MDWT module is replaced with shearlet transform. Kindly refer to section 3.1(Page number 7) of the revised manuscript.

Reviewer 2 Report

The manuscript has improved and the authors have dealt with most of the reviewer comments satisfactorily so that I can recommend acceptance. Only the following minor point should be addressed before publication:

-  Use numbers and values to support the below statement in the Abstract. “the proposed approach performs better with respect to the measures than other approaches and offers high quality fused images for an accurate diagnosis”.

Author Response

Response: As a response to the above comments, the proposed methodology is performed better results than the other state-of-art techniques and numbers, values also added that are showed in the abstract.

Reviewer 3 Report

The authors addressed all the comments. Therefore, the manuscript is acceptable in its present form.

Author Response

Response: Thank you for the positive words

Round 3

Reviewer 1 Report

The authors have successfully answered all my concerns.